# Traditional Medicinal Plants—A Possible Source of Antibacterial Activity on Respiratory Diseases Induced by *Chlamydia pneumoniae, Haemophilus influenzae, Klebsiella pneumoniae* and *Moraxella catarrhalis*

**Ligia Elena Duțu** [1], **Maria Lidia Popescu** [1], **Carmen Nicoleta Purdel** [2,*] , **Elena Iuliana Ilie** [1], **Emanuela-Alice Luță** [1] , **Liliana Costea** [1] and **Cerasela Elena Gîrd** [1]

1   Department of Pharmacognosy, Phytochemistry and Phytotherapy, Faculty of Pharmacy, "Carol Davila" University of Medicine and Pharmacy, Traian Vuia 6, 020956 Bucharest, Romania; ligia.dutu@umfcd.ro (L.E.D.); maria.popescu@umfcd.ro (M.L.P.); elena.ionita@drd.umfcd.ro (E.I.I.); emanuela.luta@drd.umfcd.ro (E.-A.L.); liliana.costea@drd.umfcd.ro (L.C.); cerasela.gird@umfcd.ro (C.E.G.)
2   Department of Toxicology, Faculty of Pharmacy, "Carol Davila" University of Medicine and Pharmacy, Traian Vuia 6, 020956 Bucharest, Romania
*   Correspondence: carmen.purdel@umfcd.ro

**Abstract:** Background. Nowadays, phytotherapy offers viable solutions in managing respiratory infections, disorders known for considerable incidence in both children and adults. In a context in which more and more people are turning to phytotherapy, finding new remedies is a topical goal of researchers in health and related fields. This paper aims to identify those traditional medicinal plants that show potentially antibacterial effects against four Gram-negative germs (*Chlamydia pneumoniae, Haemophilus influenzae, Klebsiella pneumoniae*, and *Moraxella catarrhalis*), which are considered to have high involvement in respiratory infections. Furthermore, a comparison with Romanian folk medicines was performed. Methods. An extensive review of books and databases was undertaken to identify vegetal species of interest in the context of the topic. Results. Some traditional Romanian species (such as *Mentha × piperita, Thymus vulgaris, Pinus sylvestris, Allium sativum, Allium cepa, Ocimum basilicum*, and *Lavandula angustifolia*) were identified and compared with the plants and preparations confirmed as having antibacterial effects against specific germs. Conclusions. The antibacterial effects of some traditionally used Romanian medicinal plants are poorly investigated, and deserve further attention.

**Keywords:** traditional medicinal Romanian plants; respiratory diseases; *Chlamydia pneumoniae; Haemophilus influenzae; Klebsiella pneumoniae; Moraxella catarrhalis*

## 1. Introduction

The diversity of worldwide flora, and the fact that currently, the number of people who use phytotherapy as a preventive and/or curative treatment method has an upward dynamic, are two undeniable realities [1–3].

A report from 2016 estimated that globally, the number of plant species is around 374,000 [1], of which 7.5% are used as medicinal plants [2]. At the same time, statistical reports claim that 70–95% of people continue to rely on plants as a primary form of medicine [2], and in Germany, up to 90% of the population uses herbal medicines [3].

Phytopreparations (supplements and medicines) can provide viable solutions to many health problems, including respiratory infections, one of the highest causes of death from communicable diseases. These attract the attention of health professionals through their significant incidence in children and adults. According to European Commission statistics extracted in August 2020, respiratory diseases accounted for 7.5% of deaths in the EU in 2016. Pneumonia, for example, was recorded in 2016 at the European level at a rate

of 26 deaths/100,000 inhabitants, and countries such as Portugal, Lichtenstein, and the United Kingdom had rates well above this limit [4].

Many respiratory ailments are infectious. The primary pathogens are viruses, but the involvement of bacteria and fungi should not be neglected, as they are considered direct and "co-infectious" agents [5].

The main bacteria that can cause disease in the upper and lower respiratory tract are *Streptococcus pneumoniae* (a Gram-positive germ), *Haemophilus influenzae*, and *Moraxella catarrhalis* (Gram-negative germs). In addition, other Gram-negative germs (such as *Klebsiella pneumoniae*, *Chlamydia pneumoniae*, *Coxiella burnetti*, and *Bordetella pertussis*) and Gram-positive germs (*Streptococcus pyogenes*, *Staphylococcus aureus*, and *Corynebacterium diphteriae*) have a lower involvement in terms of incidence. Still, they are essential for defining the bacterial profile of respiratory disease [6–8].

Bacterial pathogenicity involves the production of virulence factors, such as adhesins (which modulate the fixation of the pathogen on the cells of the respiratory mucosa), polysaccharides in the capsule (which help the bacterium enter the cell), exotoxins (for survival inside the host cell), or endotoxins (which induce an inflammatory, destructive response to the host cell). Of course, all these also affect the immune system of the host (human) organism. Moreover, in the case of the host having pre-existing precarious immunity issues, the pathogenicity of the bacteria is much more obvious [2].

Usually, the current treatment of bacterial infections includes antibiotics and vaccines as preventive measures. However, in many cases in managing respiratory infections, a fundamental problem is the involvement of several bacteria in the disease, each with specific structural and biochemical characteristics, certain pathogenicity, and different antibiotic resistance. In this situation, the use of phytopreparations can be a viable solution.

Plants can synthesize a significant number of secondary metabolites, which represents an effective method of combating pathogens. These metabolites have a diverse chemical structure that allows them to exert their antimicrobial effect through various mechanisms. They can act independently or synergistically or with other antibacterial agents (including antibiotics) [2,9].

The fact that phytotherapy is currently considered an essential link in modern therapy impels specialists to deepen research for medicinal plants that are already known to find new remedies. A solution in this sense may be more efficient commercialization of the indigenous flora specific to each country.

*Ethnobotany*

Ethnobotany is a multidisciplinary science that deals with the traditional knowledge of plants and their relation with people. Its practice is based on the collaboration of several researchers, such as biologists, pharmacists, physicians, anthropologists, and linguists. The aim of ethnobotanical investigation is often to select species for future pharmacological studies [10–13].

This research topic has a strong history in Romania, a country located in south-eastern Europe. Romania has a varied topography (including mountain areas, hills, and plains) and a temperate continental climate. Consequently, it has a rich and diverse flora comprising about 3700 species, of which 3100 are spontaneous species and the remainder are crop plants [14,15]. In addition, more than 700 indigenous species are known as medicinal plants and are traditionally used in various therapeutic areas, including respiratory diseases [16]. A selection of Romanian folk medicines is included in Table 1.

**Table 1.** Romanian folk medicines used in respiratory diseases and the main active compounds.

| Species (Part Used) | Main Active Compounds | Romanian Traditional Indications |
|---|---|---|
| *Allium cepa* (bulb) | Organosulfur compounds, flavonoids, and phenolcarboxilic acids [17] | Cough, pharyngitis, laryngitis, rhinitis, cold, bronchitis [11,12,18] |
| *Allium sativum* (bulb) | Organosulfur compounds, flavonoids, and phenolcarboxilic acids [17] | Cough with sputa and puss, pharyngitis, laryngitis, rhinitis, cold [11,12,18] |
| *Hyssopus officinale* (aerial part) | Polyphenols, saponins, EO (monoterpenes) [17] | Cough, laryngitis, pneumonia, tuberculosis [11,18] |
| *Juniperus communis* (shoots, berries) | Polyphenols, EO (monoterpenes) [17] | Cold, rhinitis, cough, tuberculosis [11,18] |
| *Lavandula angustifolia* (flowers) | Flavonoids, phenolcarboxilic acids, EO (monoterpenes) [17] | Cough, upper respiratory tract infections [11,18] |
| *Mentha x piperita* (leaves, aerial parts) | Flavonoids, phenolcarboxilic acids, EO (monoterpenes) [17] | Cough, asthma, pulmonary emphysema, laryngitis, tonsillitis [11,18] |
| *Ocimum basilicum* (aerial parts) | Flavonoids, phenolcarboxilic acids, EO (monoterpenes) [17] | Cough, cold, tuberculosis [11,18] |
| *Origanum vulgare* | Flavonoids, phenolcarboxilic acids, EO (monoterpenes) [17] | Asthma, pneumonia, bronchitis [11,18] |
| *Pynus sylvestris* (shoots) | Flavonoids, EO (monoterpenes) [17] | Tuberculosis, asthma [11,18] |
| *Salvia officinalis* (leaves) | Flavonoids, phenolcarboxilic acids, EO (monoterpenes) [17] | Tonsilitis, rhinitis, laryngitis, emphysema [11,12,18] |
| *Thymus vulgaris* (aerial parts) | Flavonoids, EO (monoterpenes) [17] | Cough [11] |
| *Tilia cordata* (flowers) | Mucilages, flavonoids, EO (sesquiterpenes) [19] | Cough, cold, emphysema, asthma, bronchitis, pneumonia [11,18] |
| *Verbascum phlomoides* (flowers) | Mucilages, flavonoids, iridoids [19] | Emphysema, asthma, tuberculosis, pharyngitis, pneumonia, cough, rhinitis [11,18] |

It is worth mentioning that the species named above are well-known medicinal plants all over Europe and even in China and Latin America, and are widely used in bacterial infections [20,21].

In this context, we tried to identify those traditional medicinal species with potentially antibacterial effects against four Gram-negative germs with significant involvement in respiratory infections (*Chlamydia pneumoniae, Haemophilus influenzae, Klebsiella pneumoniae* and *Moraxella catarrhalis*) [6–8]. Furthermore, a comparison with Romanian folk medicines was performed.

## 2. Materials and Methods

A literature survey was performed using a variety of Romanian ethnobotanical books [11,18] and databases, such as PubMed and Web of Science, to find the most relevant articles. Articles were limited to those published in the English and Romanian languages, focusing on the most recent works between 2010 and 2021 (75% of the cited material) but not neglecting any older relevant materials.

Initial keywords and MeSH terms were: "antimicrobial", "respiratory diseases", "plants", and "herbals". Then, the names of selected bacteria ("Chlamydia pneumoniae", "Haemophilus influenzae", "Klebsiella pneumoniae", and "Moraxella catarrhalis"), and terms related to testing antimicrobial action (such as "microbial tests", "inhibition zone", "minimum inhibitory concentration"/"MIC", and "minimum bactericidal concentration"/"MBC") were added.

In order to identify studies conducted in Romania, we used "Romania" and "Romanian" as specific keywords. At the same time, the names of some Romanian researchers (who the authors of the current work knew had studied phytochemistry and the antimicro-

bial activity of some Romanian indigenous plants) were introduced directly. Articles that existed only as an abstract and those not relevant to the proposed topic (for example, those that referred to the testing of antimicrobial action on other germs) were excluded.

In order to systematise the bibliography, the raw plant materials were finally grouped according to a geographical criterion. A total of 68 papers were selected after eligibility analysis, cross-checking and removing duplicates.

## 3. Results

The selected articles referring to testing for antibacterial effects against the four evaluated germs are systematically presented in the following tables.

Antimicrobial activity is frequently expressed as minimum inhibitory concentration (MIC), inhibition zone diameter (DIZ), or inhibition %. MIC is defined as the lowest concentration of a sample (extract, essential oil, pure compound) at which the bacteria tested did not show visible growth. Accepted criteria vary between authors. According to Fabry W. et al. [22], MIC seems to be less than 800 µg/mL for active extracts. Rios J.L. and Recio M.C. [23], Cos P. et al. [24], Kuete V. [25], Bueno J. [26], and Noundou S. [27] had more stringent criteria. They considered that an MIC equal to or less than 100 µg/mL seems to be an acceptable value to indicate a promising activity for plant extracts. On the other hand, an inactive extract is defined by Rios J.L. and Recio M.C. with a MIC above 1000 µg/mL [23]. Regarding EOs, Acs K. et al. associated a potent antimicrobial activity with an MIC value of less than 0.5 mg/mL or less than 100 µL/L [28]. For isolated compounds, Rios J.L. and Recio M.C. [23] and Kuete V. [25] considered that MIC less than 10 µg/mL suggests a high activity, and MIC above 100 µg/mL corresponds with low activity.

The diameter of the inhibition zone is defined as the diameter of the circular area around the spot of the sample (extract, essential oil, pure compound) in which bacteria colonies do not grow. Accepted criteria also vary between authors. For example, an inactive extract is associated with a diameter less 10 mm (Orbán-Gyapai O. et al.) [29], or less than 6 mm (Shihabudeen M.S. et al.) [30]. Furthermore, according to the criteria of Orbán-Gyapai O., an active extract has a 10–15 mm diameter, and a high active extract is defined with a diameter above 15 mm [29]. On the other hand, Prabuseenivasan S. et al. considered that a diameter above 7 mm indicates a promising antibacterial activity for EOs [31].

Data on *Chlamydia pneumoniae* are summarised in Table 2.

**Table 2.** Medicinal plants tested for the antimicrobial effect against *Chlamydia pneumoniae*.

| Bacterial Strain | Herbal Material/Source | Testing Sample | MIC/DIZ/Inhibition % | References |
|---|---|---|---|---|
| | Europe | | | |
| CWL-029 | *Mentha arvenisis* [R] (aerial parts)/Finland | ME | 90% at 256 µg/mL | [32] |
| K7 (clinical isolate) | *Schisandra chinensis* (fruits)/Estonia | ME | <100 µg/mL | [33] |
| | America | | | |
| AR39 | *Hydrocotyle bonariensis* (aerial parts), *Lithraea molleoides* (leaves), *Hybanthus parviflorus* (aerial parts)/Argentina | AqE, DcmE, ME Ethanol/water (1:1) extracts | 50–90% DcmE of *Hydrocotyle bonariensis* (aerial parts) is the most active | [34] |

**Table 2.** *Cont.*

| Bacterial Strain | Herbal Material/Source | Testing Sample | MIC/DIZ/Inhibition % | References |
|---|---|---|---|---|
| | Others | | | |
| K7 (clinical isolate) | 27 peppermint [R] teas /unspecified origin | AqE | 20.7–69.5% at 250 µg/mL | [35] |
| CWL-029 | Unspecified | 32 betulinic derivatives | Betulin: 53% at 1 µM Betulin-28-oxime: 100% at 1 µM Betulin-3,28-dioxime: 100% at 1 µM and 50% at 290 nM | [36] |

Abbreviations in Table 2: AqE = aqueous extract; DcmE = dichloromethane extract; ME = methanolic extract. All extracts are dry extracts. EO = essential oil. "[R]" = species identified in the Romanian flora [11,14].

Compared to the other germs included in the review, *Chlamydia pneumoniae* has been the least studiedOnly five articles were relevant, of which two refer to species of the genus *Mentha* (*M. arvensis* and *M.* × *piperita*, respectively), species also existing in the Romanian flora. *M. x piperita* is used in traditional Romanian medicine to treat respiratory infections. The MICs of the mint methanolic and ethanolic extracts are lower than 300 µg/mL.

Data on *Haemophilus influenzae* are summarised in Table 3.

**Table 3.** Medicinal plants tested for the antimicrobial effect against *Haemophilus influenzae*.

| Bacterial Strain | Herbal Material/SOURCE | Testing Sample | MIC/DIZ/Inhibition % | References |
|---|---|---|---|---|
| | Europe | | | |
| PCM2340 | *Rubus idaeus* [R] "*Willamette*"cultivar (shoots)/Poland | ME | >120 mg/mL (resistant) | [37] |
| ATCC 49247, Amp-R1, AMP-R2 | *Betula aetnensis* (leaves)/Greece | ME | 900 µg/mL (for ATCC 49247, Amp-R1), 1800 µg/mL (for Amp-R2) | [38] |
| | Africa | | | |
| clinical isolate | *Tilia cordata* [R] (bracts and flowers)/Lebanon | AqE *, ME * | 20–22 mm (flowers AqE), 0 mm (bracts AqE, AlEs) | [39] |
| ATCC 35056 | *Medicago sativa* [R] (root)/Iran | ME | 125 mg/mL | [40] |
| clinical isolates (7 strains) | *Eucalyptus globulus* (leaves)/Iran | ME * | $MIC_{50}$ = 16 mg/L, $MIC_{90}$ = 32 mg/L | [41] |
| clinical isolates | *Ammi majus* (seeds)/Oman | ME HF, CF, EaF, BF, AqF | 0 mm (ME) 6–9 mm (fractions) | [42] |
| clinical isolates (12 strains) | *Trichilia emetica* (root)/Mali | AqE DeeF | >500 µg/mL (AqE) 125 µg/mL (DeeF) | [43] |
| clinical isolates (11 strains) | 8 species of the genus *Eucalyptus* (leaves)/Tunisia | EO | 8.1 ± 2.2 mm–19.2 ± 9.6 mm | [44] |
| | America | | | |
| ATCC 49247, 90-CCH-02 | *Ceanothus coereleus* (roots), *Chrysactinia mexicana* (flowers, roots), *Cordia boissieri* (leaves), *Phyla nodiflora* (leaves), *Schinus mole* (bark, fructs, roots)/Mexico | AqE, HE, DeeE, ME | ≥500 µg/mL (all) | [45] |

**Table 3.** *Cont.*

| Bacterial Strain | Herbal Material/SOURCE | Testing Sample | MIC/DIZ/Inhibition % | References |
|---|---|---|---|---|
| Others | | | | |
| unspecified | *Echinacea. Angustifolia* [R] (root), *E. purpurea* [R] (root+aerial parts), *E. purpurea* [R] (root)/unspecified origin | Liquid AlE (48% alcohol, 40% alcohol) *; Dry AlE (0% alcohol) | AlE (40% alcohol) > AlE (48% alcohol) AlE (0% alcohol)—inactive | [46] |
| DSM 9143 | Unspecified | EO of *Syzygium aromaticum, Cinnamomum zeylanicum, Eucalyptus globulus, Thymus vulgaris* [R], *Pinus sylvestris* [R], *Mentha × piperita* [R], *Cymbopogon nardus* | By broth microdilution test: 0.25 mg/mL (*Syzygium aromaticum*), 0,06 mg/mL (*Cinnamomum zeylanicum*), 1.41 mg/mL (*Eucalyptus globulus*), 0.11 mg/mL (*Thymus vulgaris*) 1.35 mg/mL (*Pinus sylvestris*) 0.41 mg/mL (*Mentha × piperita*), 125 mg/mL (*Cymbopogon nardus*) By vapor phase test 250 μL/L (*Syzygium aromaticum*), 75 μL/L (*Cinnamomum zeylanicum*), >1500 μL/L (*Eucalyptus globulus*), 125 μL/L (*Thymus vulgaris*) >1500 μL/L (*Pinus sylvestris*) 250 μL/L (*Mentha × piperita*), 125 μL/L (*Cymbopogon nardus*) | [28] |

Abbreviations in Table 3: AqE = aqueous extract; AqF = aqueous fraction; AlE = alcoholic extract; BF = butanol fraction; CF = chloroform fraction; DeeE = diethyl ether extract; DeeE = diethyl ether fraction; EaF = ethyl acetate fraction; HE = hexane extract; HF = hexane fraction; ME = methanolic extract. All extracts are dry extracts, except those marked with "*" in table. EO = essential oil. "[R]" = species identified in the Romanian flora [11,14].

For *H. influenzae*, the number of items that met the selection criteria was slightly higher Nine referred to plant extracts (in most cases being dry aqueous or methanolic extracts) and only one to EO. MIC was evaluated in only seven articles. The investigated species come from various geographical regions (Europe, Africa, America), and nine of these are found in the Romanian flora, namely: *Tilia cordata, Thymus vulgaris, Pinus sylvestris, Mentha × piperita, Rubus idaeus, Medicago sativa, Echinacea angustifolia*, and *E. purpurea*. The first four species in this list have a long tradition of use in upper respiratory tract diseases.

The interest of researchers in finding new natural remedies for *Klebsiella pneumoniae* infection seems to be significant if we take into account the large number of articles identified in the databases and then systematized in Table 4. The anti-Klebsiella effect was investigated in different extracts (aqueous, methanolic, or ethanolic) and EO, obtained from 219 species from different regions of the globe. Twenty-two species grow spontaneously or are cultivated in Romania, and nine are relevant for the traditional Romanian phytotherapy for respiratory diseases (*Allium sativum, Hyssopus officinalis, Juniperus communis, Mentha x piperita, Ocimum basilicum, Origanum vulgare, Salvia officinalis, Verbascum phlomoides*, and *Thymus vulgaris*). The anti-Klebsiella activity is expressed as MIC, DIZ, or both.

**Table 4.** Medicinal plants tested for antimicrobial effects against *Klebsiella pneumoniae*.

| Bacterial Strain | Herbal Material/Source | Testing Sample | MIC Value/DIZ/Inhibition % | References |
|---|---|---|---|---|
| | Europe | | | |
| ATCC 700603 | 14 species of *Rumex* genus [R] (different parts)/Carpathian Basin (Hungary and Romania). | ME HE, CF, AqF | 10–15 mm (*R. acetosa, R. alpinus, R. crispus, R. aquaticys*–root CF), <10 mm (others) | [29] |
| clinical isolates | 8 aromatic plants: *Hyssopus officinalis* [R], *Achillea grandifolia*, *Achillea crithmifolia* [R], *Tanacetum partheniu* [R], *Laserpitium latifolium*, *Angelica sylvestris*, *Angelica pancicii*, *Artemisia absinthium* [R] (aerial parts)/Sebia. | ME * | 5.0 mg/mL (*Hyssopus officinalis*), 25.0 mg/mL (*Achillea grandifolia*) 2.5 mg/mL (*Achillea crithmifolia*) 25 mg/mL (*Tanacetum partheniu*) 25 mg/mL (*Laserpitium latifoliu*) 50 mg/mL (*Angelica sylvestris*) 50 mg/mL (*Angelica pancici*) 50 mg/mL (*Artemisia absinthium*) MBC > 100 mg/mL(all) | [47] |
| unspecified | 65 species from Italy flora (including *Origanum vulgare* [R]) | AlE | <4.0 µg/mL (*Origanum vulgare*) Others-inactive | [48] |
| clinical isolates | *Rubus idaeus* [R] "*Willamette*" cultivar (shoots)/Poland | ME | 60 mg/mL | [37] |
| ATCC 13882 | *Thymus vulgaris* [R] (aerial parts)/Romania | EO | 30–34 mm | [49] |
| clinical isolates (16 strains) | *Origanum vulgare* subsp. *Hirtum* [R], *Salvia officinalis* [R], *Thymus vulgaris* [R] (aerial parts)-irrigated and non-irrigated plants/Greece | EO | irrigated//non-irrigated plants: 73 mg/L//103 mg/L (*Origanum vulgare* subsp. *hirtum*) 240 mg/L//207.4 mg/L (*Salvia officinalis*) 9.5 mg/L//11.3 mg/L (*Thymus vulgaris*) | [50] |
| | Africa | | | |
| unspecified | *Medicago sativa* [R]. (seeds)/Egypt | ME * | 10 → 20 mm, depending on origin | [51] |
| clinical isolate | *Momordica charantia* (leaves and fruits)/Tanzania | ME, PeE | 12–13 mm (leaves PeE), 18 mm (fruits ME), <10 mm (others) | [52] |
| clinical isolate | *Cnestis ferruginea* (leaf)/Nigeria | AqE *, AlE *, ME * | 150 mg/mL (AqE) 20 mg/mL (AlE) 350 mg/mL (ME) | [53] |

**Table 4.** *Cont.*

| Bacterial Strain | Herbal Material/Source | Testing Sample | MIC Value/DIZ/Inhibition % | References |
|---|---|---|---|---|
| ATCC 13883 | *Warburgia salutaris* (bark, leaves)/South Africa | ME *, DcmE * | 1.0 mg/mL (bark MEs and DcmEs) 0.66 mg/mL (leaves EO) 0.50–0.83 mg/mL (bark EOs) 0.312 mg/mL (E-nerolidol) 0.13–0.208 mg/mL (other compounds) | [54] |
| ampicillin-resistant strain (unspecified) | *Curtisia dentata* (stem bark, leaves)/South Africa | ME | 156.25 μg/mL (stem bark) 312.5 μg/mL (leaves) | [55] |
| ATCC 13883 | *Xylopia aethiopica, Eriosema glomeratum,* other 16 plants (leaves)/Cameroon | Methanol-dichloromethane (1:1) extracts | 250 μg/mL (*Xylopia aethiopica*) 500 μg/mL (*Eriosema glomeratum*) 1000–> 8000 μg/mL (Others) | [56] |
| ATCC 13883 | *Alchornea cordifolia* (stems and leaves)/Cameroon | AlE, CE, EaE, ME | ≤125 μg/mL (all extracts) 16 μg/mL (Methylgallate) | [27] |
| ATCC 13883 | *Alchornea floribunda* (stems and leaves)/Cameroon | AlE, CE, EaE, ME | ≤125 μg/mL (all extracts) | [57] |
| clinical isolate | *Ocimum sanctum* (leaves), *Eugenia caryophyllata* (flowers), *Achyranthes bidentata* (stem, leaves), *Azadirachta indica* (stem and bark)/Nepal | AlE * | All: <6 mm | [58] |
| clinical isolate | *Tilia cordata* [R] (bracts and flowers)/Lebanon | AqE *, ME * | 0 mm (AqEs, MEs) | [39] |
| ATCC 700603 | *Carum copticum* (unspecified part)/Iran | AlE, ME | 12 mm (AlE), 20 mm (ME) 25 mg/mL (AlE, ME) | [59] |
| clinical isolates | *Tribulus terrestris* [R] (fruits, leaves, roots)/Iraq | AqE, AlE, CE | 0.31 mg/mL (leaves AlE) >5 mg/mL (roots AqE) 1.25. or 2.5 mg/mL (others) | [60] |
| ATCC 13883 | *Punica granatum* (leaves, flowers)/South Africa | AqE | 9–14 mm (leaves), 8–14 mm (flowers), depending on the concentration (50–5000 μL/mL) | [61] |
| ATCC 10031 | *Eucalyptus largiflorens, E. intertexta* (leaves)/Iran | ME CF, AqF EO | 10 mm (AqF), 15–20 mm (EO), 20 mm (1,8-cineol) 125 mg/mL (ME), 7.8–125 mg/mL (EO), 500 mg/mL (1,8-cineol) | [62] |

**Table 4.** *Cont.*

| clinical isolate | *Origanum syriacum, Thymus syriacus, Syzygium aromaticum, Cinnamomum zeylanicum, Laurus nobilis, Juniperus foetidissima, Allium sativum* [R], *Myristica fragrans* (leaves, bulbs, barks, aerial parts, rhizome, flowers, seeds, fruits)/Syria | AlE EO | MIC50//MIC90 6.25 µL/mL//no effect (*Origanum syriacum*) 6.25 µL/mL//no effect (*Thymus syriacus*) 1.5 µL/mL//25 µL/mL (*Syzygium aromaticum*) 3.125 µL/mL//no effect (*Cinnamomum zeylanicum*) 6.25 µL/mL//12.5 µL/mL (*Laurus nobilis*) 12.5 µL/mL//25 µL/mL (*Juniperus foetidissima*) 6.25 µL/mL//50 µL/mL (*Allium sativum*) 6.25 µL/mL//no effect (*Myristica fragrans*) | [63] |
|---|---|---|---|---|
| unspecified | *Ocimum basilicum* [R] (aerial parts)/Oman | EO | resistant | [64] |
| unspecified | *Thymus capitatus* (aerial parts)/Libya | EO | 4.0–5.0 mm | [65] |
| unspecified | *Salvia lachnocalyx*, S. *mirzayanii* and *S. sahendica* (aerial parts)/Iran | EO | 10 mm | [66] |
| NCTC 9633 | *Artemisia afra* (leaves and stems), *Agathosma betulina* (leaves), *Eucalyptus globulus* (leaves), *Osmitopsis asteriscoides* (leaves)/South Africa | EO | 9.3 mg/mL (*Artemisia afra*—leaves and stem) 16 mg/mL (*Agathosma betulina*-leaves), 8.0 mg/mL (*Eucalyptus globulus*-leaves), 8.0 mg/mL (*Osmitopsis asteriscoides*-leaves) | [67] |
| clinical isolates (13 strains) | 8 species of the genus *Eucalyptus* (leaves)/Tunisia | EO | 6.6–10.8 mm, depending on origin | [44] |
| ATCC 13883 | *Leptospermum petersonii, L. scoparium, Kunzea ericoides* (aerial parts)/South Africa | EO | 8.0 mg/mL (all EOs) | [68] |
| | | Asia | | |
| ATCC 4352 | *Anethum graveolens* [R] (aerial parts, leaves, seeds)/Turkey | Aerial parts EO Seed EO | 3.13%–12.5% (aerial parts EO), 0.8–12.5% (seed EO) | [69] |
| ATCC 4352 | *Verbascum xanthophoeniceum, V. densiflorum* [R], *V. lagurus V. gnaphalodes, V. phlomoides* [R] (aerial parts)/Turkey | ME * CF *, EaF *, PeF *, TF*, AqF * | 312.5 µg/mL (*V. lagurus* EaF) >1250 µg/mL (others) | [70] |

**Table 4.** *Cont.*

| | | | | |
|---|---|---|---|---|
| unspecified | 5 medicinal plants: *Boerhaavia diffusa, Cassiaauriculata, Cassia lantana, Eclipta alba, Tinospora cardiofolia* (leaves)/India | AqE ME | *Boerhaavia diffusa:* 10 mm (AqE, ME), *Cassia auriculata, Cassia Lantana:* 0 mm (AqE, ME), *Eclipta alba:* 9 mm (AqE), 16 mm (AlE),*Tinospora cardiofolia:* 8 mm (AqE), 13 mm (AlE) | [71] |
| unspecified | 6 folk medicinal plants in India: *Eugenia jambolana* (kernel), *Cassia auriculata* (flowers) *Murraya koenigii* (leaves), *Salvadora persica* (stem) and *Ipomoea batatas* (leaves) and *Andrographis paniculata* (leaves)/India. | ME | *Andrographis paniculate:* 8 mm (at 2 mg/mL)–12 mm (at 4 mg/mL) *Eugenia jambolana:* 7 mm (at 2 mg/mL)–12 mm (at 4 mg/mL) *Cassia auriculata:* 9 mm (at 2 mg/mL)-12 mm (at 4 mg/mL) Other species and its extracts: <6 mm | [30] |
| ATCC 10273 | *Acacia melifera* (whole plant)/India | HE *, EaE *, ME *, AlE * | 18 mm (ME), 0 mm (other extracts) | [72] |
| ATCC 4030 | *Alysicarpus vaginalis* (root)/India | AqE, CE, ME, PeE | 10 mm (AqE, PeE), 11 mm (CE), 12 mm (ME) 6.25 mg/mL (ME) | [73] |
| NCIM2719 | 23 species belonging to 21 different families (leaves, stem)/India | AcE, ME | 8–21 mm | [74] |
| unspecified | *Pogostemon benghalensis* (leaves)/India | AqE, ME | 0 mm (AqE–cold water), 8–12 mm (AqE-hot water), <8 mm (ME-cold methanol), >12 mm (ME-hot methanol) | [75] |
| clinical isolates | 26 ayurvedic plants (different parts)/Bangladesh | Extracts (unclear specified) | 8–21 mm, depending on species and bacterial isolates: 10–17 mm (*Allium sativum*-bulb), 9–13 mm (*Allium cepa*-bulb), 8–10 mm (*Nigella sativa*-seeds), 9–21 mm (*Citrus limonum*–fruits) s.a. | [76] |

**Table 4.** *Cont.*

| | | | | |
|---|---|---|---|---|
| clinical isolates | *Coriandrum sativum* [R] (seeds), *Brassica alba* [R] (seeds), *Mentha arvensis* [R] (leaves), *Ocimum basilicum* [R] (leaves), *Terminalia bellirica* (fruits), *Illicium verum* (fruits), *Hyptis suaveolens* (seeds), *Vetiveria zizanioides* (roots), *Myristica fragrans* (fruits), *Sesamum indicum* (seeds), *Piper nigrum* (fruits), *Curcuma longa* (rhizome)/Bangladesh | AlE, EaE, HE | 187.5 μg/mL (*Coriandrum sativum* seeds AlE) 375 μg/mL (*Brassica alba* seeds EaE and HE) 750 μg/mL (*Mentha arvensis* leaves AlE) 750 μg/mL (*Ocimum basilicum* leaves HE) 187.5 μg/mL (*Terminalia bellirica* fruits EaE), 93.7 μg/mL (*Terminalia bellirica* fruits AlE) 375 μg/mL (*Illicium verum* fruits EaE and HE), 1500 μg/mL (*Illicium verum* fruits EaE) 375 μg/mL (*Hyptis suaveolens* seeds HE) >1500 μg/mL (*Vetiveria zizanioides* roots HE) 375 μg/mL (*Myristica fragrans* fruits HE), 1500 μg/mL (*Myristica fragrans* fruits AlE and EaE) 375 μg/mL (*Sesamum indicum* seeds HE) 375 μg/mL (*Piper nigrum* fruits HE and EaE) 375 μg/mL (*Curcuma longa* rhizomes HE) | [77] |
| ATTC 13883 | 6 non-indigenous medicinal plants naturalized in Malaysia: *Ailanthus triphysa, Clinacanthus nutans, Gynostemma pentaphyllum, Gynura bicolor, Turnera subulata* (leaves), *Asystasia gangetica* (aerial parts) | AlE, AqE, CE, EaE, HE, ME | AqEs–inactive 2.5 mg/mL (*Ailanthus triphysa* leaves AlE and ME), 2.5 mg/mL (*Clinacanthus nutans* leaves AlE); *Clinacanthus nutans* leaves ME-inactive, 1.25 mg/mL (*Gynostemma pentaphyllum* AlE); *Gynostemma pentaphyllum* ME-inactive, 1.25 mg/mL (*Gynura bicolor* AlE); *Gynura bicolor* ME-inactive, 2.5 mg/mL (*Turnera subulata* leaves AlE and ME) 1.25 mg/mL (*Asystasia gangetica* aerial parts AlE), 0.63 mg/mL (*Asystasia gangetica* aerial parts ME) | [78] |
| MTCC-432 | *Ocimum basilicum* [R] (leaves)/India | EO | 15 μg/mL | [79] |
| NCIM 2957 | *Ocimum basilicum* [R] (flowering aerial parts)/India | EO | MBC = 1.875 mg/mL | [80] |

**Table 4.** *Cont.*

| ATCC 15380 | Unspecified | EOs/India (market) | 3.2 mg/mL (Cinnamom EO)<br>>6.4 mg/mL (Clove EO)<br>12.8 mg/mL (Geranium EO, Orange EO)<br>>12.8 mg/mL (Lemon EO, Rosemary EO) | [31] |
|---|---|---|---|---|
| | | America | | |
| unspecified | *Ocimum basilicum* [R] (leaves, stems and flowers)/Brazil | EO | 12.2 mm<br>0.75 mg/mL | [81] |
| | | Others | | |
| ATCC 13883 | *Coriandrum sativum* [R] (seeds)/unspecified origin | EO | 0.2% | [82] |
| ATCC 700603 | Unspecified | EOs | 5 mg/mL (Peppermint [R] EO)<br>20 mg/mL (Eucalyptus EO)<br>10 mg/mL (Cajuput EO, Wintergreen EO)<br>40 mg/mL (Juniper [R] Berry EO) | [83] |
| unspecified | *Allium sativum* [R] (bulbs)/(unclear specified) | AqE, CE, EaE, HE, ME | 12 mm (ME), 17 mm (AqE), <10 mm (others)<br>150 µg/mL (ME), 100 µg/mL (AqE) | [84] |
| ATCC 700603 | Unspecified | Curcumin | 216 µg/mL | [85] |

Abbreviations in Table 4: AcE = acetone extract; AqE = aqueous extract; AqF = aqueous fraction; AlE = alcoholic extract; BE = butanol extract; CE = chloroform extract; CF = chloroform fraction; DcmE= dichloromethane extract; EaE = ethyl acetate extract; EaF = ethyl acetate fraction; HE = hexane extract; HF = hexane fraction; ME = methanolic extract; PeE = petroleum ether extract; PeF = petroleum ether fraction; TF = toluene fraction. All extracts are dry extracts, except those marked with "*" in table. EO = essential oil. "R" = species identified in the Romanian flora [11,14].

Data obtained on *Moraxella catarrhalis* are summarised in Table 5.

Regarding *Moraxella catharrhalis* we identified 20 articles that met the applied selection criteria, of which fifteen referred to plant extracts and four to EO. Only one investigated the cumulative anti-Moraxella effect of EO, methanolic, and dichloromethane extracts of *Warburgia salutaris* bark. Among the 72 investigated species, eleven can be found in the Romanian flora: *Rubus idaeus*, *Medicago sativa*, *Allium sativum*, *Mentha x piperita*, *Pinus sylvestris*, and *Thymus vulgaris*, together *with* five species of the genus *Rumex*. The first four species also attract attention by their association in various traditional preparations used in the treatment of respiratory diseases. The majority of the authors considered MIC the most relevant parameter for the purpose, compared to DIZ.

**Table 5.** Medicinal plants tested for the antimicrobial effect against *Moraxella catarrhalis*.

| Bacterial Strain | Herbal Material/Source | Testing Sample | MIC value/DIZ/inhibition % | References |
|---|---|---|---|---|
| | | Europa | | |
| ATCC 25238 | *Betula aetnensis* (leaves)/Greece | ME | 220 µg/mL | [38] |
| ATCC 43617 | 14 species of *Rumex* genus [R] (different parts)/Carpathian Basin (Hungary and Romania). | ME<br>AqF, CF, HF | *R. aquaticus* [R] (roots, aerial parts), *R. crispus* [R] (aerial parts), *R. patienta* [R] (flowers), *R. stenophyllus* [R] (flowers), *R. thypsiflorus*[R] (roots):<br>>10 mm (AqF),<br>others: <10 mm | [29] |

**Table 5.** *Cont.*

| Bacterial Strain | Herbal Material/Source | Testing Sample | MIC value/DIZ/inhibition % | References |
|---|---|---|---|---|
| ATCC 43617 | 4 bryophyte species (*Amblistegium serpens, Plagiomnium cuspidatum, Rhytidium rugosum, Schistidium crassipilum*)/Hungary | AqE, ME CF, HF | 9.0 mm (*Amblistegium serpens* CF) 10.0 mm (*Plagiomnium cuspidatum* HF) 7.5 mm (*Rhytidium rugosum* CF) 7.7 mm (*Schistidium crassipilum* CF) | [86] |
| ATCC 25238 | *Thymus capitatus* (leaves)/Italy | ME HF, MF | 62.5 µg/mL (MF) >1000 µg/mL (others) | [87] |
| ATCC 25238 | *Helleborus bocconei* subsp. *siculus* (root)/Italy | ME | 0.4 mg/mL | [88] |
| PCM 2340 | *Rubus idaeus* [R] (3 cultivars), *Rubus occidentalis* (1 cultivar) (fruits)/Poland | AlE | 2–8 mg/mL (extracts) 0.015 mg/mL (elagic acid) | [89] |
| PCM2340 | *Rubus idaeus* [R] "Willamette" cultivar (shoots)/Poland | ME | 0.5 mg/mL | [37] |
| Africa | | | | |
| ATCC 23246 | *Warburgia salutaris* (bark)/South Africa | DcmE *, ME * EO | 0.42 mg/mL (DcmE), 2.0. mg/mL (ME) 0.5–1.0 mg/mL (EO) 0.031 mg/mL (E-nerolidol) | [54] |
| ATCC 25240 | *Punica granatum* (leaves, flowers)/South Africa | AqE * | 8.9–14 mm (leaves), 12.0–15.33 mm (flowers); depending to the concentration (50–5000 µL/mL) | [61] |
| ATCC25240 | *Medicago sativa* [R] (root)/Iran | ME | 16 mm 125 mg/mL | [40] |
| ATCC 23246 | *Artemisia afra* (leaves and stems), *Agathosma betulina* (leaves), *Eucalyptus globulus* (leaves), *Osmitopsis asteriscoides* (leaves)/South Africa | EO | 8 mg/mL (all) | [67] |
| ATCC 23246 | *Leptospermum petersonii, L. scoparium, Kunzea ericoides* (aerial parts)/South Africa | EO | 4 mg/mL (*Leptospermum petersonii*), 2 mg/mL (*L. scoparium*), 8 mg/mL (*Kunzea ericoides*) | [68] |
| ATCC 23246 | *Citrus limon* (leaves), *Eucalyptus grandis* (leaves), *Helichrysum kraussii* (leaves and stem), *Lippia javanica* (leaves), *Tetradenia riparia* (leaves)/South Africa | EO | 13.33 mg/mL (*Citrus limon*-leaves), 4.0 mg/mL (*Eucalyptus grandis*-leaves), 5.33 mg/mL (*Helichrysum kraussii*-leaves and stem), 5.33 mg/mL (*Lippia javanica*-leaves), 5.33 mg/mL (*Tetradenia riparia*-leaves) | [90] |
| ATCC 14468 | 18 species *Alchornea floribunda* (leaves) *Musanga cecropioides* (leaves and stem bark) *Tetracera potatoria, Xylopia aethiopica* (stem barks)/South Africa | Methanol— dichloromethane (1:1) extracts | 65 µg/mL (*Alchornea floribunda*-leaves), 130 µg/mL (*Musanga cecropioides*-leaves and stem bark) 250 µg/mL (*Tetracera potatoria*–stem bark), 250 µg/mL (*Xylopia aethiopica*-stem bark) >1000 µg/mL (others) | [56] |
| clinical isolates | *Trichilia emetica* (root)/Mali | AqE EeF | >500 µg/mL (AqE), 7.8–31.2 µg/mL (EeF) | [43] |

**Table 5.** *Cont.*

| Bacterial Strain | Herbal Material/Source | Testing Sample | MIC value/DIZ/inhibition % | References |
|---|---|---|---|---|
| clinical isolates | *Allium sativum* [R] (bulbs), *Cinnamomum zeylanicum* (bark), *Syzygium aromaticum* (buds), *Persea americana* (leaves), *Rosmarinus officinalis* (leaves), *Argemone mexicana* (leaves)/Ethiopia | AlE *, AqE *, ME * | 15.0 mm (*Allium sativum* AqE), 11.0 mm (*Cinnamomum zeylanicum* AlE), 11.0 mm (*Persea americana* ME), no inhibition (others). 30 mg/mL (*Allium sativum* AqE), 20 mg/mL (*Cinnamomum zeylanicum* AlE), 30 mg/mL (*Persea americana* ME), no inhibition (others) | [91] |
| unspecified | *Curtisia dentata* (leaves)/South Africa | AcE, AlE, CE, EaE | 6.25 mg/mL (AlE), 1.57 mg/mL (CE), 3.13 mg/mL (AcE and EaE) 0.3 mg/mL (betulinic acid) 1.25 mg/mL (ursolic acid) 3.13 mg/mL (lupeol) 1.25 mg/mL (β-sitosterol) | [92] |
| ATCC 23246, | *Alchornea cordifolia* (roots, stems and leaves)/Cameroon | AqE, AlE, CE, HE, ME | 125 µg/mL (AlEs and MEs) >1000 µg/mL(others) | [27] |
| ATCC 23246 | *Alchornea floribunda* (roots, stems and leaves)/Cameroon | AqE, AlE, CE, HE, ME | 250 µg/mL (roots AlEs and MEs) 500 µg/mL (leaves AlEs and MEs) 1000 µg/mL (stems AlEs and MEs) | [57] |
| Others | | | | |
| DSM 9143 | Unspecified | EOs of *Syzygium aromaticum, Cinnamomum zeylanicum, Eucalyptus globulus, Thymus vulgaris* [R], *Pinus sylvestris* [R], *Mentha × piperita* [R], *Cymbopogon nardus* | By broth microdilution 0.25 mg/mL (*Syzygium aromaticum*) 0.10 mg/mL (*Cinnamomum zeylanicum*) 2.81 mg/mL (*Eucalyptus globulus*), 0.09 mg/mL (*Thymus vulgaris*) 1.34 mg/mL (*Pinus sylvestris*) 0.35 mg/mL (*Mentha × piperita*) 0.11 mg/mL (*Cymbopogon nardus*) By vapor phase test 125 µL/L (*Syzygium aromaticum*) 25 µL/L (*Cinnamomum zeylanicum*) 225 µL/L (*Eucalyptus globulus*) 50 µL/L (*Thymus vulgaris*) >1500 µL/L (*Pinus sylvestris*) 31.25 µL/L (*Mentha × piperita*) 25 µL/L (*Cymbopogon nardus*) | [28] |

Abbreviations in Table 5: AcE = acetone extract; AqE = aqueous extract; AqF = aqueous fraction; AlE = alcoholic extract; CE = chloroform extract; CF = chloroform fraction; DcmE = dichloromethane extract; DeeE = diethyl ether fraction; EaE = ethyl acetate extract; HF= hexane fraction; HE = hexane extract; ME= methanolic extract. All extracts are dry extracts, except those marked with "*" in table. EO = essential oil. "[R]" = species identified in the Romanian flora [11,14].

## 4. Discussion

Numerous species from the spontaneous and cultivated flora of different countries have been studied for their antibacterial activity against the following four germs: *C. pneumoniae, H. influenzae, K. pneumoniae,* and *M. catarrhalis*. Only a few published studies investigated herbs from the Romanian flora.

The antimicrobial effect is frequently expressed as MIC, DIZ, or inhibition %. However, DIZ is an obsolete criterion because its value may vary depending on multiple factors, such as sample solubility and concentration, the thickness of the agar medium or the rate of drug diffusion through agar. Therefore, the MIC value is considered a more suitable and reliable antibacterial criterion, and we used this parameter in our assessment.

- Plant-based antibacterial agents active on *Chlamydia pneumoniae*

*C. pneumoniae* belongs to the genus *Chlamydia* and is a Gram-negative bacteria without any known animal reservoir, which spreads via respiratory droplets and induces pneumonia. It is an obligate intracellular bacterial pathogen that infects the respiratory tract. *C.*

*pneumoniae* undergoes a biphasic life cycle, alternating between a smaller extracellular form, the elementary body (EB), and a larger replicating intracellular form, the reticulate body (RB). The infectious EB attaches to susceptible host cells and enters cells via endocytosis, inhibiting phagolysosome fusion. Then EB matures into a noninfectious RB that is separated from the cytosol within nonlysosomal inclusions [93]. Inside these inclusion bodies, *C. pneumoniae* creates an intracellular niche, whereby it can modify host cell pathways, replicate, and revert to the EB form before cell lysis [94].

*C. pneumoniae* initially infects lung epithelial cells and alveolar macrophages. Alveolar macrophages secrete significantly higher amounts of IL-1β and lower amounts of IL-1R-antagonist [95]. Furthermore, the production of IL-1β is stimulated by NOD-like Receptor (NLR) family member NLRP3, which detects cellular stress induced by *C. pneumoniae* infection [96].

Several *C. pneumoniae* antigens have been implicated in activating the innate immune response. For example, in a *C. pneumoniae* lung infection, both Toll-like receptor (TLR) 2 and TLR4 use the MyD88 (the myeloid differentiation primary response 88) pathway to recognize chlamydial components, such as lipopolysaccharide or chlamydial heat shock protein 60 (cHSP60). However, TLR2 plays a significant role in host responses to *C. pneumoniae* infection by producing inflammatory cytokines that activate the cell-mediated immune response, predominantly T-helper (Th)1 [97]. *Chlamydia* infection increases the release of inflammatory cytokines within the alveoli, resulting in local destructive effects on lung tissue. Although *C. pneumoniae* infection is predominantly asymptomatic or mild, it can lead to acute upper and lower respiratory illnesses, including bronchitis, pharyngitis, sinusitis, and pneumonia [98]. It is estimated that 5–10% of cases of bronchitis, pharyngitis, sinusitis and pneumonia have Chlamydia pneumoniae as the pathogen. Some studies have identified that prolonged treatment with azithromycin, clarithromycin, or levofloxacin might lead to phenotypic resistance to these antibiotics [99].

Several studies demonstrated the anti-Chlamydia effect of herbal preparations). Some of them are derived from species that exist in Romanian flora, such as *Mentha* sp., *Medicago sativa* L., *Trifolium pratense* L., or *Betula* sp.

The genus *Mentha* includes over 25 species and hybrids, spontaneous or cultivated. Their large pharmacologic spectrum (including antimicrobial and anti-spasmodic effects on respiratory tract) is based on complex chemical composition, species-dependent variables including phenol carboxylic acids, flavonoids (flavones, catechols), and EO rich in monoterpenes [17,100].

Regarding the anti-Chlamydia effect of mint, there are studies worldwide. For example, Kapp K. et al. highlighted excellent results with different aqueous extracts at a concentration of 250 μg/mL. It has been suggested that the effect may be due to flavonoids, such as luteolin and apigenin glycosides [35].

In another in vitro study, using one standard strain (CWL-029) and one clinical strain (K7) of *C. pneumoniae*, Salin O. compared the effect of a hydroalcoholic extract of *Mentha arvensis* L. (cornmint) with that of the isolated compounds (linarin and rosmarinic acid). The concentration of 256 μg/mL of the hydroalcoholic dry extract inhibited the formation of chlamydial inclusion by more than 60% without influencing the level of intracellular ATP. The antichlamydial effect exhibited by the isolated compounds linarin and rosmarinic acid, which are also the main compounds quantified in the extract, demonstrates their involvement in the antibacterial effect of the extract. The result was also confirmed in vivo, using C57BL/6I inbred mice inoculated with K7 culture as biological material. Furthermore, the hydroalcoholic extract and the isolated compounds decreased some inflammatory parameters associated with *C. pneumoniae* infection (such as IgG antibody levels), and may prevent the long-term side effects of diseases with this pathogen [32].

Yanazaki T. demonstrated the antimicrobial effect of catechols from a product called "Polyphenon" (a tannoid tea leaf complex with 18.3% epigallocatechin, a representative compound for mint species) against two Chlamydia strains (AR-39 and AR-43) [100]. Based on the results, and considering that polyphenols have good bioavailability through topical

administration [101], the authors propose to develop a preparation that should be used as inhalation therapy in respiratory infections caused by *C. pneumoniae* [102].

In addition, to potentiate the anti-Chlamydial effect, Salin O. et al. propose the association of polyphenolic compounds, such as quercetin, luteolin, and octyl gallate, with synthetic calcium channel blockers (such as isradipine, verapamil, or thapsigargin). However, they also demonstrated that combination with doxycycline does not improve the effectiveness of the antibiotic. On the contrary, at high doses, there is also the risk of antagonistic effects [9].

In Romanian flora, *Mentha arvensis* and *Mentha x piperita* are present [14], but unfortunately we could not identify studies on the anti-Chlamydia effect of Romanian spontaneous or cultivated *Mentha* species in the consulted databases.

Nevertheless, the presence of phenolic compounds as flavonoids (for example, 1.91–3.37 mg% luteolin) and the GC/MS chromatographic profile of Romanian mint essential oil [103–105] indicate a possible future capitalization of the Romanian mint in the treatment of *C. pneumoniae* infection.

Species of the genus *Betula* are hardwood tree species in the flora of Europe, and are phytochemically characterized by the presence in the leaf of flavonoidic compounds and volatile sesquiterpenes, and lupanic-type triterpenes (including betulin, betulinic acid, esters, and glycosidic derivates) in the bark, respectively. Traditionally, in many European countries the birch leaf (collected from *Betula pendula* Roth.) is used only to treat urinary disorders. The bark is currently being studied as a source of betulin derivatives, potential antibacterial agents in respiratory disorders [106,107].

To our knowledge, there are no studies on *Betula pendula* Roth. leaves.

In contrast, Salin O. et al. studied 32 synthetic betulin derivatives in an in vitro acute infection model using TR-FIA (Time-resolved fluorometric immunoassay method). They concluded that the % of inhibition varied depending on the chemical structure of the derivates. For example, betulin inhibited 53%, betulinic acid only 19%, and betulinic acid esters had much less or no inhibition, while the oximes and dioximes were very active, inhibiting over 95%. Among these, betulin dioxime was the most active compound [36].

There are five species of the genus *Betula* [14] in Romania, of which *Betula pendula* Roth. leaves are traditionally used as diuretic. As triterpenes lupeol-type were also identified in samples of *Betula pendula* bark from Romania [108], this species could be further investigated for its anti-Chlamydia effect.

*Medicago sativa* L. (alfalfa) and *Trifolium pratense* L. (red clove) are known for their value as medicinal plants [109–111]. Alfalfa aerial-part extracts and red clover-flower extracts contain isoflavones, but their anti-Chlamydia effect has not yet been studied. However, Hanski L. et al. reported inhibitory effects of isolated isoflavones, such as biochanin A, genistein, formonetin, daidzein, and daidzin against Chlamydia. Of these, biochanin A, a methylated isoflavone, is the most active [112].

*Medicago sativa* L. and *Trifolium pratense* L. are cultivated in Romania [14,113,114]. Moreover, isoflavones have been identified in other species exist in the Romanian flora, e.g., *Genista tinctoria* L., *Glycyrrhiza glabra* L., *Glycyrrhiza echinata* L., *Ononis spinosa* L., *Genistella sagitalis* L., *Cytisus albus* Hacq., *Coronilla varia* L., *Lotus cornyculatus* L., and *Dorycnium herbaceum* Vill. [115].

Considering all aspects, we can hypothesize that all these species from Romanian flora, rich in isoflavones, could inherit an anti-Chlamydial effect.

- Plant-based antibacterial agents active on *Haemophilus influenzae*

*H. influenzae* is a Gram-negative pleromorphic coccobacillus. It is found in humans in the nasopharynx and throat, represents a significant cause of meningitis in children, and causes upper and lower respiratory tract infections in children and adults [116].

Some strains of *H. influenzae* are encapsulated, while others are non-encapsulated. Six antigenic serotypes (designated a–f) of encapsulated *H. influenzae* based on their capsular polysaccharide were identified. The major virulence factor is the polyribose-phosphate capsule of type b H influenzae [117].

Both encapsulated and non-capsulated strains of *H. influenzae* can cause respiratory tract infections. However, encapsulated *H. influenzae* is the common cause of invasive *H. influenzae* infection, including pneumonia in young children. In contrast, non-capsulated strains of *H. influenzae* are generally considered a significant cause of chronic respiratory disease and pneumonia in adults.

The clinical manifestations of the lower respiratory tract infection include bronchitis, which may be acute or chronic, bronchiectasis, and cystic fibrosis. Additionally, *H. influenzae* is also a significant cause of pneumonia and acute otitis media [117]. In most cases of pneumonia, there is multilobular, maculate, diffuse, and usually bilateral involvement of the pulmonary tissue. The mortality rate for *H. influenzae* pneumonia ranges between 30% and 40% [118].

*H. influenzae* acute otitis media occurs more commonly as bilateral disease, with slight fever or pain and frequently associated eye symptoms, so-called "otitis–conjunctivitis syndrome" [119].

A proportion of *H. influenzae* isolates produce β-lactamase, while Ampicillin-resistant non-β-lactamase strains are prevalent in Japan and Spain [120,121].

In the context in which the identification and development of new active antibiotics on ampicillin-resistant *Haemophilus influenzae* is a medium priority of the WHO [122], phytotherapy can have an important role.

There are numerous studies about the antibacterial effect of different extracts against *H. influenza*), including some species that are found also in the Romanian flora (e.g., *Tiliae* sp., *Medicago sativa* L., *Betula* sp., *Echinacea* sp., *Thymus vulgaris* L., *Helleborus* sp., *Mentha* × *piperita* L., and *Pinus sylvestris* L.) [14,123].

Ismail A. demonstrated the antibacterial action of *Tilia cordata* Mill. (linden) against *H. influenza* for a methanolic extract and flower infusion, harvested from Lebanon linden specimens, while the bract dry extracts were inactive. The author assumed that the difference in effect was due to the different chemical compositions [39]. In inflorescence, flavonoids and mucilages predominate, while catechic tannins can be found in bracts [19,39].

Studies on the chemical composition of linden flowers from Romania are poorly represented. Mircea C. et al. comparatively analyzed two commercial samples of linden flowers and reported a content of 479–647 mg flavonoids (expressed as rutin)/100 g dry sample and 663–1169 mg polyphenols (expressed as caffeic acid)/100 g dry sample) [124].

Currently, the scientific literature does not provide enough data to demonstrate a similarity of chemical composition between the linden flowers from Lebanon and those from Romania. Therefore, this partially limits appreciation of the anti-Haemophilus effect of the Romanian species.

*Medicago sativa* L. (alfalfa), a member of the *Fabaceae* family, exhibits antibacterial properties against *H. influenzae*. In this regard, Chegini H. et al. demonstrated that at an MIC of 1.25 μg/mL, the alcoholic dry extract of the root of alfalfa produces significant inhibition of the pathogen. However, this effect is lower than the one against *M. catarrhalis* and *Streptococcus pneumoniae* [40]. Future phytochemical and microbiological studies are necessary to demonstrate the anti-Haemophilus effect of the Romanian alfalfa species.

*Betula leaf* (birch) is a well-known antibacterial agent [19,106]. Acquaviva B. et al. carried out a study on an extract (qualitative and quantitative composition is not specified) from the leaf of *Betula aetnensis*, an endemic species from the eastern slope of Etna. The authors demonstrated that the standard *H. influenzae* strain ATCC 49247 and the ampicillin-resistant Amp-R1 are susceptible to this extract (MIC was 900 mg/mL and 1800 mg/mL, respectively) [38].

In Romanian flora, *Betula pendula* is present [14]. Costea T. et al. reported the presence of phenolic compounds (caffeic acid, chlorogenic acid, ferulic acid and p-coumaric acid) and flavonoids (quercetol, myricetol, apigenol and kaempferol as free aglycones and heterosides) in leaves of *Betula* species collected from Arges county (Romania) [125]. These compounds were also found in a similar *Betula pendula* leaf sample from Italy [126]. As a

result of these data, we can appreciate that preparations from Romanian species deserve to be investigated in the future for their anti-Haemophilus effect.

*Helleborus* species are herbaceous species found in the temperate zone in Europe. Their chemical composition includes cardiotonic glicosides bufadienolidic-type (e.g., hellebrin), steroidal saponins, ecdisteroids, and gamma-lactones, as protoanemonin [127]. In Romanian folk medicine, the root is used to treat respiratory infections in pigs and sheep [128].

Puglisi S. et al. published the results of a study performed on a methanolic extract from the root of *Helleborus bocconei* Ten. subsp. *siculus*, an endemic species in central and southern Italy. They showed a moderate anti-Haemophilus effect of the total extract obtained from the root, lower than that recorded for the bufadienolidic fraction. Compared to the impact against other germs studied and involved in respiratory infections (*Streptococcus pneumoniae*, *M. cathartis*), the effect of the methanolic extract on Haemophilus was lower [88].

Four species belonging to the *Helleborus* genus are identified in Romanian flora [14]. However, only *Helleborus purpurascens* W. et al. has aroused some interest. The research focused on developing a method for the extraction, identification and assay of hellebrine [129], and the antioxidative effect assessment of a selective fraction obtained using a preparative chromatographic technique [130].

The presence of hellebrin in *Helleborus bocconei* Ten. subsp. *siculus* from Italy and *Helleborus purpurascens* from Romania, correlated with the anti-Haemophilus action of the first-mentioned species, opens the possibility of future research of the Romanian species.

*Echinacea* species are well known in therapy due to their immunostimulatory effects. In Europe, food supplements and phytomedicines containing extracts of root and/or aerial parts of *Echinacea angustifolia* D.C., *E. purpurea* (L.) Moench. and *E. pallida* Nutt. are recommended to treat colds and flu. These diseases have in the causal sphere both infection with various bacterial and viral pathogens, and deficiency of the immune system [131,132].

Sharma M. et al. compared the intensity of the anti-Haemophilus effect induced by six extracts derived from *E. angustifolia* (root) and *E. purpurea* (root and aerial parts) extracts previously characterized in terms of the content of polyphenols (expressed as caffeic acid), alkylamides, and polysaccharides. The authors have shown that extracts with moderate content of alkylamides and caffeic derivatives and without polysaccharides are more active against this germ. In contrast, extracts of *E. purpurea*, characterized by high polysaccharide content and medium content of caffeic derivatives and alkylamides are inactive. Based on the results, the authors could not attribute the antibacterial effect to one of the classes of monitored compounds [46].

In Romania there are cultures of *Echinacea*, the root and the herb being commercialized as dietary supplements. In 2018 there were 58 notified supplements manufactured in Romania or imported, 52% of which are monocomponent and 29% of which contain other herbs, extracts, or vitamins, while 19% are registered as herbal tea [133].

Phytochemical studies of samples from Romania have shown the presence of compounds characteristic of *Echinacea* species, previously mentioned and involved in therapeutic effects (including antibacterial effect). Using TLC and HPLC techniques, Elek F. et al. characterized 12 commercial samples of mono- or multi-component supplements in terms of the content of phenolic acids (caffeic acid, caftaric acid, cichoric acid, and chlorogenic acid) and echinacosides, thus being able to identify plant raw material/producing species [133].

Furthermore, the study conducted by Banica F. et al. should be mentioned. Using two different methods of analysis (spectrometry and Differential Pulse Voltammetry), the authors established the content of polyphenols and the polyphenolic profile (caffeic acid, caftaric acid, cichicoric, and catechols) of three samples of nutritional supplements containing extracts of *E. purpurea*, and correlated the phytochemical results with their antioxidative activity [134].

To date, there are no studies on the antibacterial effect of some Echinacea extracts of Romanian origin against *H. influenzae*. Still, the existence in the Romanian samples of

some active principles cited by Sharma M. may be an argument for initiating research in the future.

EO are complex mixtures of volatile monoterpenes, sesquiterpenes, aromatic compounds, and rarely, diterpenes. From a phytochemical point of view, the EO is characterized by its chromatographic fingerprint, a parameter that also influences the pharmacological profile. Furthermore, the amount and the qualitative and quantitative composition of EO depend on several intra- and interspecific variables, such as the identity of the species, the plant part used, the pedoclimatic conditions in which the species grow, applied agro-technical measures (for crops), primary processing measures, or the experimental extraction conditions [17,47,50,69,135,136]. As a result, more than in "non-volatile" medicinal herbals, in aromatic species, the chemical composition of an EO varies qualitatively and quantitatively within wide limits, which also generates differences in pharmacological and toxicological profiles (such as type and intensity of effects). The anti-Haemophilus effect is not an exception.

In Table 3 are listed some EOs that are active against *H. influenzae*, including thyme EO, peppermint EO, and pine EO.

Acs K. et al. analyzed anti-Haemoplilus effect for EOs obtained from *Mentha x piperita* L. (peppermint), *Thymus vulgaris* L. (thyme), and *Pinus sylvestris* L. (scots pine) (cultures from Hungary, a geographically neighboring country with Romania), in two experimental models—in the liquid phase (broth microdilutions method) and the vapor phase, respectively. The SHS-SPME-GS-MS analysis highlighted as majority constituents: in peppermint EO-menthol (27.2%), menthone (19.8%), izomenthone (11.4%), and eucalyptol (17.4%); in thyme EO-thymol (46.1%), ϒ-terpinen (6.5%), and p-cymen (27.9%); in scots pine EO–α-pinen (26.1%), β-pinen (18.0%), limonen (17.0%), and ϒ-3-caren (14%). while phellandrene was not identified. Thyme and mint EOs have proven to be much more active than pine EO, especially in the liquid phase, as evidenced by the results: in the broth microdilution method, MIC was 0.11 mg/mL for thyme EO, 0.21 mg/mL for peppermint EO, and 1.35 mg/mL for scots pine EO, and in the vapor phase test, MIC was 25 μL/mL for thyme EO, 50 μL/mL for peppermint EO, and 500 μL/mL for scots pine EO. According to the criteria of Acs K. [28], thyme EO and mint EO could be anti-H. influenzae agents, while scots pine EO is inactive. The stronger effect in the liquid phase could be due to direct contact with the pathogen, altered membrane permeability, and degradation of cell morphology [28]. For at least the peppermint EO, the cytotoxic activity can be explained based on the content of terpenes that interact with biomembranes in a non-covalent manner, increasing their fluidity and permeability [136,137]. When scanning databases, no studies about the antibacterial effect on *H. influenzae* of peppermint, thyme, and pine EOs with Romanian origin were found. However, there are some reports about the chemical profiles. The GS/MS analysis of some EO samples from Romanian cultures of *Mentha × piperita palescens* revealed the following composition: 39.69% menthol, 15.74% menthone, 7.73% izomenthone, 1.5% eucalyptol, 2.1% piperitone, and 2% pulegone. The same authors also analyzed two other species that inhabit Romania [14]: *Mentha spicata* L. (whose essential oil contains 12.77% menthol, 41.21% carvone, and 7% menthone) and *Mentha suaveolens* Ehrh. (with 73.77% piperiton-oxide, 0.128% menthol, 0.3% piperitone, and 1.5% carvone in EO) [104].

Regarding thyme EO, Boruga O. et al. reported 47.59% thymol, 30.90% ϒ-terpinen, and 8.41% p-cymen in a sample from Mehedinti county (southwestern Romania) [49], while Aprotosoaie A.C. et al. sustained a content of 55.44% thymol, 5.74% ϒ-terpinen, and lack of p-cymen in a sample from the Moldavian zone (Eastern Romania) [138]. Their report also mentioned three other spontaneous species in the Romanian flora [14], whose volatile oil was chromatographically characterized: *Thymus pulegium* L. from Prahova country (with 1.6–6.6% thymol, 50.5–62.6% carvacrol, 5.8–7.0% p-cymene, 9.8–9.9% ϒ-terpinene, and geraniol absent), *Thymus glabrescens* L. also collected from the mountain zone of Prahova country (with 1.5% thymol, 4.7% carvacrol, and 55.0% geraniol) [49] and *Thymus dacicus* from Gorj country (with 5.397% thymol, 0.365% carvacrol, 18.376% geraniol, 7.466% p-cymene, and 0.278% ϒ-terpinene) [139].

To our knowledge, only Swiss pine EO (from Pinus cembra inhabit in Romanian flora) was studied phytochemically. The twig EO contained a much higher amount of a mixture of limonene and phellandrene (40.0%) and a higher content of $\alpha$-pinene (24.94%) and $\beta$-pinene (10.38%), but much lower content of $\Upsilon$-3-carene (1.03%), than the oil analyzed by Acs K. [28,140].

Considering the existence of common chemical constituents in concentrations that are sometimes comparable, we believe that thyme, peppermint, and scots pine EOs extracted from native species in Romania may be the subjects of future research to outline a possible antibacterial effect against *H. influenzae*.

- Plant-based antibacterial agents active on *Klebsiella pneumoniae*

*K. pneumoniae*, which belongs to the *Enterobacteriaceae* family, is a rod-shaped, Gram-negative, encapsulated, non-motile bacillus. It is considered an opportunistic, hypervirulent, and multidrug-resistant pathogen [141]. It has been associated with pneumonia, especially in critically ill and immunocompromised patients. Nowadays, *K. pneumoniae* pneumonia is considered the most common nosocomial infection, ranging from mild to severe [142].

*K. pneumoniae* typically colonizes human mucosal surfaces, including the oropharynx and gastrointestinal tract. The colonization rates of the nasopharynx range from 3% up to 15%, and are higher in adults than in children [143]. The primary established virulence factors are the polysaccharide capsule that protects the bacteria from phagocytosis, the secretion of multiple types of siderophores, and the pili that help to adhere to the cell surface [144]. In addition, efflux pump AcrAB and a type VI secretion system have also been identified as virulence factors [141]. After entering the body, *K. pneumoniae* subverts efferocytic uptake by neutrophils and activates necroptosis of infected neutrophils [145].

Pneumonia induced by *K. pneumoniae* usually affects the upper lobes. Clinical examination reveals unilateral signs of consolidation, such as crepitation, bronchial breathing, and increased vocal resonance, in the upper lobe. A hallmark of infection with *K. pneumoniae* is the "currant jelly" sputum due to necrosis of the surrounding tissue.

*K. pneumoniae* often display a high rate of antibiotic resistance, including carbapenem resistance, making it difficult to choose appropriate antibiotics for treatment [146]. Due to the high resistance, the development of new active antibiotics on K. pneumoniae is a critical WHO priority [122].

Medicinal plants were tested as anti-Klebsiella agents, of which some are also present in the Romanian flora, e.g., *Nigella sativa* L., *Rubus idaeaus* L., *Allium sativum* L., *Alllium cepa* L., *Origanum vulgare* L., *Coriandrum sativum* L., *Artemisia absinthium* L., *Ocimum basilicum* L., and *Echinacea* sp. [14].

*Nigella sativa* L. (black cumin) is a herbaceus species belonging *Ranunculaceae* family with a complex chemical composition that includes among others: EO (rich in thymoquinone and derivatives), fatty oil (unsaturated fatty acids are major components), fitosterols, [147,148], polyphenols (sinapinic acid, ferulic acid and derivatives, kaempferol, and quercetol) [149]. Various extracts and the seed EO are currently used in therapy, including for respiratory disorders [147,148].

*Nigella* species are intensively phytochemical and pharmacologically studied, including for their action against *K. pneumoniae*. For example, in a study using ten cultures of Klebsiella isolates, Chowdburry M.A.N. et al. concluded that *Nigella sativa* extract has a moderate antibacterial activity, lower than those exhibited by two other extracts analyzed in parallel (extracts of fruit of *Citrus limonum* and *Tamarindus indica*) [76].

In Romania, incipient research on the chemical composition of indigenous species (*Nigella sativa* L. and *N. damascena* L.) was carried out by Toma C.C. et al., but they refer only to the identification and dosing of polyphenols and flavonoids in alcoholic extracts (in ethanol 70%, *v/v*) [150], which is insufficient to guarantee a similar behavior to the sample tested by Chowdburry M.A.N. et al. in terms of antibacterial effect.

*Rubus ideaus* L. (raspberry) is a cultivated *Rosaceae*, known primarily for the nutrition value of its fruits. In therapy, the fruits are used in the treatment of digestive disease

due to their anti-inflammatory and antiseptic properties. The compounds of interest are anthocyanins and hydrolyzed tannins (ellagitannins), e.g., ellagic acid and sanguiin H-6 [151]. At the same time, in many European countries, raspberry shoots are used for infections and inflammation of the upper respiratory tract [152].

Krauze-Boranowska M. et al. reported that *K. pneumoniae* is a germ sensitive to a methanolic dry extract of raspberry shoots of the "*Willamette*" cultivar. HPLC-DAD-ESI-MS analysis allowed the characterization of the methanolic dry extract, namely: phenolic acids (gallic acid, caffeic acid, chlorogenic acid, catechic acid, ellagic acid and derivatives predominates), flavonoids (quercetol, kaempferol, myricetin, and their corresponding oxygen-glycosides), catechols, and proantocyanidins B1 and B2. Of these, the majority compounds are ellagic acid (5256.0 ± 400.5 mg/100 g extract) and sanguiin H (1151.7 ± 102.9 mg/100 g extract), compounds suggested to be involved in antibacterial activity, but without any adequate evidence in this regard [37].

Any studies attesting the presence of these compounds in Romanian raspberry samples or regarding the testing of the anti-Klebsiella action were not identified. By using an HPLC-UV method, Costea T. et al. identified and assayed other phenolic compounds (caffeic acid, ferulic acid, and p-coumaric acid) and flavonoids (quercitrin, izoquercitrin, quercetin, and kaempferol) in raspberry leaves collected from Ilfov county flora (Romania) [153]. Given that the detection was performed by comparing the UV spectra in the sample with those of the available reference substances, other compounds (e.g., ellagic acid and sanguiin-4) may be present in sufficient quantity to impress the anti-Klebsiella effect. Therefore, the *Rubus idaeaus* leaf should be included on the list of herbs that deserve to be tested for possible anti-Klebsiella effects.

Apart from its culinary use, *Allium sativum* L. (garlic) is a medicinal plant. Organosulfur compounds, flavonoids, and phenolcarboxilic acids are the main compounds involved in the antioxidative, antibacterial, and anti-inflammatory effects of garlic bulb [154,155].

Regarding its anti-Klebsiella action, according to Chowdburry A. report, based on the evaluation of DIZ in the culture of 10 Klebsiella isolates, that this is a moderate but superior activity compared to that recorded for *Nigella sativa* [76].

In another study, Meriga B. et al. showed the superior effect of dry aqueous garlic extract compared to methanolic dry extract, as evidenced by the values recorded for MIC (100 μg/mL and 150 μg/mL, respectively). They partially attributed the effect to sulfur compounds (e.g., allicin), which, on the one hand, are involved in the production of messenger-RNA and RNA-transfer (and so inhibits protein synthesis), and on the other hand, in the lipid synthesis (so the microbial cell wall is damaged) [84]. The involvement of sulfur compounds is also supported by the subsequent study carried out by Al-Mariri A. on the EO extracted from the garlic bulb (MIC$_{90}$ = 50 μL/mL) [63].

According to the information provided by the accessed databases, the garlic cultivated in Romania has been the object only of qualitative phytochemical studies. Using the HPLC method, Pârvu M. identified sulfur compounds (alliin and alliicin) and polyphenols (gentisic acid, chlorogenic acid, 4-hydroxibenzoic acid, and p-coumaric acid) in a sample from Cluj county extracted with 20% ethanol (*w/v*) [156]. Meanwhile, Trifunschi S. hypothesized the presence of phenolic compounds in a sample from Arad county extracted with 50% ethanol, based on the characteristic peaks recorded by IR spectrometry [157]. The presence of sulfur compounds and phenolic acids in samples from Romania, similar to those tested for antimicrobial effect on *K. pneumoniae*, recommends further phytochemical research on Romanian garlic to quantify these active principles and, depending on the result, test specific antimicrobial action.

Another species of the genus, *Allium cepa* L. (onion), is considered both a food and a medicinal plant with antibacterial properties [158]. However, its anti-Klebsiella potential is slightly lower than that manifested by *Allium sativum* [76]. According to the criteria of Orbán-Gyapai O., these extracts have a moderate antibacterial activity [29].

Studying the effect on some *Klebsiella* isolates of the total dry extract and of the corresponding fractions obtained from it (using as solvents petroleum ether, chloroform, ethyl

acetate, and butanol), Bakht J. demonstrated a higher potency for the apolar dry fractions, the most active being chloroform fraction, which caused 62% inhibition compared to the positive control of ciprofloxacin [159]. Unfortunately, there is no information about the anti-Klebsiella effect of onion extract grown in Romania, but only phytochemical data. For example, Tataringa G. et al. identified in the onion bulb volatile compounds trisulphides (propenyl, as well as propyl trisulphide as the majority compound) and disulphydes (dipropyl disulphide, bis (1-methylethyl) disulphide, and 1-methylethyl propyl disulphide) [160], while Oancea S. characterized the anthocyanidin fraction of two red onion cultivars [161]. Considering the above-presented studies, onions that grow in Romania can be a potential anti-Klebsiella agent.

*Tribulus terestris* L. is a spontaneous herbaceous species belonging to *Zygophylaceae* family, widespread in Europe (including Romania) [14]. It is well-known in therapy as a diuretic, aphrodisiac, and hormone booster in men and women, and is used to treat impotence, infertility, and urinary disorders [162–164].

Al-Bayati A.F. and Al-Mola H. studied the antibacterial effect of aqueous, ethanolic, and chloroform dry extracts obtained from the root, leaf, and fruit of *Tribulus terestris* on 11 bacterial strains (including *K. pneumoniae*). The preparations were obtained using collected specimens from Iraq. The leaf ethanolic extract proved to be more active (MIC = 0.31 mg/mL) than that from the fruits (MIC = 1.25 mg/mL), superior even to Maxipine, cephalosporin taken as reference (MIC = 0.62 mg/mL). The researchers suggested that sterol saponosides, flavonoid glycosides, phytosterols, amides, and alkaloids are involved in the antibacterial effect [60].

Accessed databases did not provide information on the chemical composition and antibacterial effect of *Tribulus terestris* extracts obtained from specimens collected from Romania. Stefanescu R. identified dioscin in fruit samples marketed in Romania, but it is unclear if the plant source was from Romania [163]. Dioscin, protodioscin, prototribetin, and rutin, active principles cited by Al-Bayati, were also identified in samples from Bulgaria, a country bordering Romania [165]. As a result, *Tribulus terestris* collected from the Romanian flora may be an active antibacterial agent against *K. pneumoniae*.

*Origanum vulgare* L. is a representative of *Lamiaceae* family. It is known in phytotherapy for its antiseptic properties (on germs also located at the respiratory tract), These actions involve phenolic acids (rosmarinic acid) and flavonoids (heterosides of luteolin and apigenin), as well as volatile terpenes in EO (thymol, carvacrol etc.) [17,166]. Azzo A.A. studied the effect of 68 dry ethanolic extracts (80% ethanol as extraction solvent) against *K. pneumoniae*. Of these, only *Origanum vulgare* was active (MIC < 4 μg/mL), while others, such as *Marrubium vulgare* (aerial parts), *Sambucus nigra* (flowers), and *Thymus serpyllum* (flowering tops), were inactive in tested concentrations (4–12 μg/mL) [48]. All species mentioned above can be found in Romanian flora [14] and are known for their positive effects in treating respiratory ailments [17,166–169].

Although we do not know if there are studies on the anti-Klebsiella action of some plant product samples collected from the Romanian flora, we consider that the research should not be abandoned. The negative results reported by Azzo A.A. can be justified considering: (1) the variability of the chemical composition depending on the pedoclimatic conditions; (2) possible selective effect on other germs located at respiratory level; (3) inadequate extraction method used (solvent, extraction parameters); (4) insufficient concentration used in the microbiological test.

In another study, Hossan Md.S. reported the effects of 54 dry extracts obtained from 18 herbals by extraction with various solvents (hexane, ethyl acetate, ethanol). Extracts of *Coriandrum sativum* (coriander) fruits, *Mentha arvensis* (corn mint) leaves, and *Ocimum basilicum* (basil) leaves were inactive (MIC > 100 μg/mL) [77]). However, these species are known for their antibacterial and anti-inflammatory effects on the respiratory tract [100,170–173]. The unexpected results may be due to the type of extract used or the test conditions, or perhaps these extracts are active on other germs but not on *K. pneumoniae*.

Unfortunately, there are no anti-Klebsiella studies for extracts of Romanian origin, even in this case. However, we can point out the phytochemical analysis by Trifan A. et al. on coriander fruits. Using the HPLC-DAD-ESI-Q-TOF-MS/MS technique, they identified chlorogenic acid, dicaffeoilquinic acid, luteolin, and apigenin-pentoside in the methanolic extract. Luteolin-C-hexozide, apigenin-C-hexozide-O-pentoside, apigenin-C-hexozide, quercetin-C-hexozide, and kaempferol-O-hexozide [174], as active compounds, were also cited by other researchers [173].

As a result, we consider that phytochemical research should be extended for Romanian coriander fruits and, depending on the results, discussion should continue regarding the potential for microbiological studies.

*Artemisia absinthium* L. (wormwood) is a member of *Asteraceae* family. The chemical composition of its aerial parts contains flavonoids (quercitin and rhamnetin and their glycosides), phenolic acids (coumaric acid and chlorogenic acid), terpenes guaianolidic-type (absinthins), volatil terpenes (*trans*-thyjone, myrcene, and 1,4-terpinen). Digestive disorders are the main therapeutic domain in which wormwood is used [175].

At the same time, some studies have shown its antibacterial effect, including on *K. pneumoniae*. For example, Stankovich N. et al. analyzed the effect of a methanolic wormwood dry extract on Klebsiella sputum-isolated strains. The results obtained prove a moderate effect (MIC = 50 mg/mL; MBC = 100 mg/mL). Since the MBC/MIC ratio is less than 4, a bactericidal effect may be suggested [44]. Unfortunately, the study contains only limited phytochemical information. The researchers determined only the content of polyphenols (80–120 mg polyphenols, expressed in gallic acid equivalents/100 g extract) and the content of flavones (60–80 mg flavonoids, expressed as rutin equivalents/100 g extract) [47].

*Artemisia absinthium* is also found in the Romanian flora [14], and it has been the subject of some phytochemical research. For a leaf ethanol extract, Craciunescu O. et al. reported a content of 179 mg/g polyphenols (expressed as caffeic acid) and 52 mg/g flavone (expressed as quercetin), both calculated at an extraction yield of 14.28%. In addition, HPLC was used to identify and assay the flavonoid compounds (quercetin was the major compound $-2707$ mg/g, along with luteolin, apigenin and myricetin $-0.677$ mg/g, 0.399 mg/g and 0.201 mg/g, respectively) and phenolic acids (caffeic acid 0.181 mg/g) [176]. Ivanescu O. et al. used HPLC/MS analysis to characterize a methanolic dry extract of wormwood herb. They highlighted the presence of phenolic acids (caffeic, chlorogenic, p-coumaric), and flavonoids (quercetin, apigenin, rutosides, and hyperosides) [177]. Moaca E.A. et al. compared wormwood stem and leaf in ethanolic solutions and concluded that the two extracts had a similar thermogravimetric and FT-IR phenolic profile [178]. To our knowledge, so far, the product from Romania has not been microbiologically tested for its anti-Klebsiella action.

Several EOs are effective against *K. pneumoniae*. These include basil and coriander EOs.

The EO extracted from flowering aerial parts of *Ocimum basilicum L.* (a culture from Brasil), having 71.01% linalool, was active against a standard Klebsiella strain (MIC = 0.75 mg/mL) [81]. In contrast, Al-Abbasy D.W. et al. claimed that *K. pneumoniae* is resistant to basil EO, containing 69.86% linalool [64]. In another study, Joshi R.K. showed a weak effect of the EO obtained from Indian basil herb (MBC = 1.875 ± 0.684 mg/mL). According to GC/MS report, methyl-eugenol (39.3%), and methyl-chavicol (38.3%) were the main compounds [80].

Two samples of Romanian basil EO, corresponding to two geographical areas, were studied by Benedec D. Monoterpenes (linalool, 1,8-cineole) are the majority compounds in the Dolj county sample (49.15%). In contrast, the sample's Cluj county is rich in sesquiterpenes (52.97% epi-bicyclosesquiphellandrene, cadinene, farnesene, and elemene) [179]. Sesquiterpenes proved to be the majority in a sample tested by Andro A.R., among which the quantified compounds were δ-cadinol (18.21%), germacren D (17.18%), β-cadinene (12.34%), Υ-cadinene (7.36%), and α-bergamotene (7.18%) [180]. The origin of the plant

is not explicitly specified. Still, from the context of the article, it can be assumed that it is from Romania.

Regarding coriander EO, the microbiological study performed by Silva F. should be noted. He found a moderate effect (MIC = MBC = 0.2 mg/mL) [82], which could be attributed especially to linalool [181]. The presence of linalool as a major constituent of EOs has also been reported for coriander fruits from various European countries, the concentration of linalool ranging from 58.0% to 80.3%, depending on the country of origin. In these oils, linalool is accompanied by other terpene compounds, such as: Υ-terpinene (0.3–11.2%), α-pinene (0.0–10.9%), limonene (0.1–3.2%) [182].

Unfortunately, as with many other herbal drug and EOs, there are no studies on the anti-Klebsiella effect of Romanian coriander oil. However, it is well-known that a similar content of major compounds may induce a comparable effect. So, it is worth noting the phytochemical study conducted by Tsaghi A. et al. for a sample from Harghita county. The results (48.4% linalool, as the main compound, 9.2–12.1% Υ-terpinene, 5.5–9.3% α-pinene, 4.7–6.3% limonene) [183] suggest a possible microbiological behavior similar to that reported by Silva S. [82].

A significantly increased content of linalool (18.46–39.5%) and linalyl acetate (25.64–29.86%) in the flowers of *Lavandula angustifolia* cultivated in the Transylvania region of Romania [184] suggests that this species also deserves to be the subject of microbiological research for assessment of the anti-Klebsiella effect.

Other plant extracts and volatile oils were also tested to evaluate the sensitivity of the Gram-negative bacterium *K. pneumoniae*, e.g., *Tilia cordata* Mill., *Rubus idaeaus* L., *Echinacea sp.*, *Mentha piperita* L., *Brasica nigra* L., *Angelica sylvestris*, and *Trigonella goenum-graecum* L.

The methanolic extract from linden flowers (*Tilia cordata*) significantly inhibited the growth of the bacterium when ratios of extract volume to agar volume were less at 0.10 and 0.05, and can also inhibit biofilm formation [39]. However, because there are no data to prove at least a similar phytochemical profile, we cannot comment on the effectiveness of the Romanian preparations.

Other extracts, such as alcoholic and aqueous extracts obtained from *Echinacea* root and/or herb of *Echinacea angustifolia* and *E. purpurea* [46], alcoholic and acetone dry extracts of *Trigonella foenum-graecum* seeds [74], hexane dry extracts, ethyl acetate dry extracts of *Brasica nigra* seeds and *Mentha arvensis* leaves, and alcoholic and aqueous extracts of *Tribulus terestris* root and fruits) [60] proved to have poor activity.

*K. pneumoniae* was resistant to the dry methanolic leaf extract of *Rubus ideaus* [89], as well as to the Eos of *Mentha piperita* [63], *Rosmarinus officinalis* [63], *Brassica nigra* [63].

All the above species are present in the Romanian flora [14]. The lack of data to demonstrate the phytochemical similarities of the Romanian plants to the species proven as anti-Klebsiella agents requires a reserved approach to forming conclusions on the antimicrobial action of Romanian species against *K. pneumoniae*.

- Plant-based antibacterial agents active on *Moraxella catarrhalis*

*M. catarrhalis* is a pathogen Gram-negative non-capsulated diplococcus. Almost 20% of acute bacterial otitis media in children and nearly one-third of exacerbations of chronic obstructive pulmonary disease symptoms in adults are caused by this pathogen [185]. It is also a well-known cause of community-acquired pneumonia.

The genetic analysis indicates that *M. catarrhalis* comprises two distinct lineages that differ in their potential for virulence [186].

*M. catarrhalis* can colonize the mucosal surfaces, including bronchial epithelial cells, small airway epithelial cells, and type 2 alveolar cells [187]. *M. catarrhalis* is also present intracellularly in human pharyngeal lymphoid tissue, thus suggesting a potential reservoir for persistence in the human respiratory tract [188]. The adhesion to surfaces is a multifactorial event mediated by many *M. catarrhalis* adhesin macromolecules.

Furthermore, the expression of several proteins, such as integral outer membrane proteins including the ubiquitous surface proteins A1 and A2 [185], immunoglobulin D

binding protein [189], filamentous hemagglutinin-like proteins, and *M. catarrhalis* porin-like outer membrane protein CD, contribute to *M. catarrhalis* adhesion to epithelial cells [190].

Once attached to the host mucosal surfaces, *M. catarrhalis* can form microcolonies and biofilms, and subverts innate host immune responses. In addition, *M. catarrhalis* interferes with the classical pathway of the complement system by binding to C4b-binding protein and C3-protein using UspA1 and A2 [191].

Pneumonia induced by *M. catarrhalis* is characterized by productive cough, progressive dyspnea, and fever. Chest radiography reveals patchy infiltrates and occasional lobar consolidation.

The clinical symptoms of chronic obstructive pulmonary disease exacerbation due to *M. catarrhalis* are comparable to those induced by *H. influenzae* and *S. pneumoniae*, and include increased sputum production, sputum purulence, and dyspnea [192].

More than 90% of *M. catarrhalis* produce β-lactamases, and are thus resistant to ampicillin [186].

The studies selected from the accessed databases on herbals tested for their anti-Moraxella effect are summarized above.

Romanian flora can offer solutions in this microbiological direction through species such as *Allium sativum* L., *Medicago sativa* L., *Rubus idaeus* L., *Rosmarinus officinalis* L., *Thymus vulgaris* L., *Mentha* sp., and *Pinus* sp. [14]. The mentioned species are also active against germs previously discussed (*C. pneumoniae*, *H. influenzae*, and *K. pneumoniae*). Still, the intensity of the effect varies depending on the tested germ, the botanical variety, and the plant part used as a consequence of different chemical compositions. Some examples are mentioned below.

Using two clinical *M. catarrhalis* isolates with different sensitivity to antibiotics, Rasheed M.U. et al. compared the effect of five extracts. These were obtained from garlic (*Allium sativum*) bulb, cinnamon (*Cinnamomum zeylanicum*) bark, clove (*Syzygium aromaticum*) flowers, rosemary (*Rosmarinus officinalis*) leaf, avocado (*Persea americana*) leaf, and prickly poppy (*Argemone mexicana*) leaf. Garlic extract proved to be most active on both *Moraxellla* strains, followed by cinnamon and avocado extracts. In contrast, rosemary and prickly poppy extracts were inactive. Thus, the antibacterial effect of garlic may be due, at least in part, to allicin, as evidenced by the decrease in the intensity of the effect by heating [91].

Chegini H. et al. reported promising results from a *Medicago sativa* root methanolic dry extract on five germs involved in sinusitis and bronchitis. Against *M. catarrhalis*, at a concentration of 125 mg/mL, the methanolic extract induced a higher inhibition than on *Streptococcus pneumoniae* and *H. influenzae* [40]. According to the criteria of Orbán-Gyapai O., alfalfa seems to be a highly active extract.

The anti-Moraxella action of the methanolic extract from the shoots of *Rubus idaeus* cultivar *Willamette* is superior (MIC = 0.47 mg/mL) to that recorded on *K. pneumoniae* (MIC = 60 mg/mL) and on *H. influenzae* [37]. The same extract (from shoots) showed a stronger effect than the extract from the fruits of *Ribes idaeus* cultivars *Ljulin* (MIC = 4.0 mg/mL), *Veten* (MIC = 4.0 mg/mL), and *Porrana Rosa* (MIC = 2.0 mg/mL) [89].

As K. et al. demonstrated, of the 7 EOs studied (including thyme, mint, and pine), the thyme EO proved to be the most active, its action being also superior to that manifested on *H. influenzae* [28].

Studies on the anti-Moraxella effect of Romanian herbals were not identified. Still, the presence of similar constituents to those identified in samples that exhibited anti-Moraxella activity may be an argument for the initiation of further microbiological studies.

- Retrospective discussions

A retrospective evaluation of the bibliographic data identified more than 250 species that were tested against *C. pneumoniae*, *H. influenzae*, *K. pneumoniae*, and *M. catarrhalis*. Thirty-five of them are found in Romanian flora, but also in many other countries. There are no studies to certify anti-Chlamydia, anti-Haemophilus, anti-Klebsiella, or anti-Moraxella effects in samples from the Romanian flora; only samples collected/harvested from other countries were used, However, it should be mentioned that some of these species are cited

in Romanian folk medicine for the treatment of respiratory diseases [11,12,18] (Table 1). From a phytochemical point of view [17,19], polyphenols (including flavonoids and phenolcarboxilic acids) and volatile monoterpenes seem to be the main active compounds in these plants. Having in view only MIC results of Romanian species, some extracts and EOs seem to be active against the four microbes. According to Rios J.L. criteria [22,23], of 17 species cited in the database, methanolic dry extracts of *Mentha arvensis* and *Mentha* × *piperita* seem to be active against *Chlamydia pneumoniae*, and methanolic dry extract of *Rubus idaeaus* is active on *Moraxella catarrhalis*. The anti-Klebsiella agents list is more extended, including alcoholic extracts of *Tribulus terestris*, *Coriandrum sativum*, *Mentha arvensis*, *Ocimum basilicum*, *Allium sativum* and *Origanum vulgare*, and aqueous extract of *Allium sativum*. On the other hand, according to Acs K. criteria [28], thyme EO and peppermint EO have promising activity against *Haemophilus influenzae*. Thyme EO is also active against *Moraxella catarrhalis*, and basil EO is a potent anti-Klebsiella agent. The extrapolation of the reported results for the extracts obtained on raw materials with another origin is debatable. Geoclimatic conditions significantly influence the chemical composition and influence the species pharmacology [2,47,193].

Generating a scale that correlates the intensity of antimicrobial effect on a specific germ based on comparative analysis of the existing published data is difficult, due to the coexistence of several variables. These variables can be classified as: (1) related to the tested material (plant source, the extraction solvent used, the extraction technique) and (2) related to the antimicrobial effect, such as the description of the conditions for testing the antimicrobial action and the type of parameter investigated. Regarding the first category, it is unanimously accepted that the solvent has a major influence on extract composition and implicitly on its pharmacology and toxicity. In many cases, the extract toxicity is also due to solvent toxicity, limiting its use in phytotherapy. As listed in previous tables, some authors microbiologically tested aqueous and alcoholic extracts, which can be directly used in therapy. At the same time, there are many studies in which the solvent used for extraction is a solvent class 2 (chloroform, dichloromethane, hexane, methanol, or toluene) [194]. Due to residual solvent content, for these extracts, the microbiological results have only secondary/preliminary support for their valorization in therapy, even if they are dry extracts. To our knowledge, so far, there are no phytopreparations including these types of extracts in the market.

Other variables that must be mentioned: the analytical method used to characterize the extract (active compounds monitored, method of analysis, the method of reporting results), the methodology for antimicrobial action (type of method, concentrations used, sample processing, experimental test conditions, antibiotic used as a positive control), and the parameters that are taken into account for the assessment of the antimicrobial activity (DIZ, MIC, MBC, inhibition %). In addition, some articles contain only results for antibacterial testing without giving information about the source and chemical profile of the analyzed extracts.

At the same time, insufficient data (or even lack thereof) on the plant source, the extraction solvent used, and the description of the conditions for testing the antimicrobial action further limit the value of the results of some studies.

## 5. Conclusions

The published data confirm the antibacterial effects of some preparations against the Gram-negative germs *C. pneumoniae, H. influenzae, K. pneumoniae,* and *M. catarrhalis,* but none of these preparations were derived from Romanian samples. Nevertheless, considering the results of studies performed on samples with similar composition and the limitations mentioned above, it can be hypothesized that some species collected from the Romanian flora may be regarded as new therapeutic solutions in treating respiratory infections. The potential species included on the list are: *Allium cepa* L., *Allim sativum* L., *Lavandula angustifolia* L., *Mentha piperita* L., *Ocimum basilicum* L., *Pinus sylvestris* L., and *Thymus vulgaris* L.

However, this list is indicative only. Further studies on samples collected from Romania are needed to confirm their antimicrobial effect against these four significant germs and possibly to discover new and safer alternatives to antibiotics.

**Author Contributions:** Conceptualization, L.E.D. and C.E.G.; Data curation, L.E.D., M.L.P., C.N.P., E.I.I., E.-A.L., L.C. and C.E.G.; Investigation, L.E.D., M.L.P., E.I.I., E.-A.L., L.C. and C.E.G.; Methodology, L.E.D. Project administration, L.E.D.; Supervision, L.E.D.; Visualization, L.E.D., C.N.P. and C.E.G.; Writing—original draft, L.E.D. and C.N.P.; Writing—review & editing, L.E.D. and C.N.P. All authors have read and agreed to the published version of the manuscript.

**Funding:** This research received no external funding.

**Institutional Review Board Statement:** Not applicable.

**Data Availability Statement:** Not applicable.

**Conflicts of Interest:** The authors declare no conflict of interest.

## Abbreviations

| | |
|---|---|
| *C. pneumoniae* | *Chlamydia pneumoniae* |
| *H. influenzae* | *Haemophilus influenzae* |
| *K. pneumoniae* | *Klebsiella pneumoniae* |
| *M. catarrhalis* | *Moraxella catarrhalis* |
| DIZ | diameter of the inhibition zone |
| MIC | minimum inhibitory concentration |
| MBC | minimum bactericidal concentration |
| EO | essential oil |

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
