# Peer review of "Traditional Medicinal Plants—A Possible Source of Antibacterial Activity on Respiratory Diseases Induced by Chlamydia pneumoniae, Haemophilus influenzae, Klebsiella pneumoniae and Moraxella catarrhalis"

_diversity, doi:10.3390/d14020145_

Round 1

Author Response

Dear reviewer, thank you for taking the time to read this manuscript. Your feedback is appreciated. Please find attached a document with our answers to your comments/suggestions and the revised manuscript.

Reviewer 2 Report

The title: please use italic in names of microbes. It's too extensive (Romanian flora). Respiraotry diseases are not mentioned in the titel.

Page 1, line 37-39 (The diversity of worldwide flora and the fact that currently, the number of people 37 who use phytotherapy as a preventive and/or curative treatment method has an upward 38 dynamic are two undeniable realities.) Reference? How the authors know that?

Page 2, line 44-45 (Phytopreparations (supplements and medicines) can provide viable solutions to 44 many health problems, including respiratory diseases. These attract attention to health 45 professionals through the significant incidence in children and adults.) Why only respiratory diseases are mentioned?

Page 3, line 97 (A literature survey was performed using PubMed, Web of Science, Google Scholar.) Why not also Scopus which goes more and more important database? From the other side, Google Scholar is too general.

Result: there are only long Tables without any comments. Also interpretation of Tables needed. Partially, the results are described in the chapter Discussion.

Discussion: this section is too long containing a lot of unimportant details.

Generally: it is not clear why the authors of the manuscript emphasize the Romanian flora. The species they describe are growing also in lot of other countries. The manuscript is too much orientated only to Romania. On the same time we do not have sufficient information about chemotypes of medicinal plants in Romania and in other regions.

Author Response

Dear reviewer, Thank you for taking the time to read this manuscript. Your feedback is appreciated. Please find attached a document with our answers to your comments/suggestions and the revised manuscript.

Reviewer 3 Report

Dear authors,

The title of your manuscript suggests a report on Romanian plant species signalized by the inhibitory activity of their herbal preparations against those bacteria. This is not the case. In fact the manuscript is a review on the inhibitory activity of herbal preparations from plant species of worldwide origin having in common investigations and results on the inhibitory activity of their extracts or fractions against the most prevalent bacteria associated to respiratory diseases. Followed by an exercise of identification of which of those plant species also grow wild or are cultivated in Romania, Then a simple and disappointing conclusion was formulated - "these plant species deserve further attention" !!!

Under a different title that reflected the real contents of the review, the manuscript could be seen as consistent, supported on a suitable search of literature, reporting relevant and detailed information on the target microorganisms, results on their inhibition by the plant preparations and isolated compounds. It is focused on a topic that is rarely object of attention and it can be of interest of some readers. Nevertheless, in such case I´m not sure if it falls within the aims of the special issue "Ethnobotany, Medicinal Plants and Biodiversity Conservation": the review and the most of the cited papers have not support on ethnobotany or traditional herb knowledge, do not discuss on therapeutic efficacy of the mentioned medicinal plants and extracts for respiratory infectious diseases, are preliminary in regard any active compound potentially useful to inspire new medicines, nor on preservation of wild populations and r sustainable use.

In conclusion, and assuming that editors will consider that the manuscript falls within the aim of the special issue, I recommend a major revision asking you:

- to adapt the manuscript (title, discussion and conclusion) in accordance to the objective contents of this review, antibacterial activity of plant preparations on the most prevalent bacteria associated to respiratory diseases: Chlamydia pneumoniae, Haemophilus influenzae, Klebsiella pneumoniae, and Moraxella catarrhalis.;

Avoid extrapolate conclusions in regard to the flora of a particular region, or in regard to species other then those who were investigated;

- following the discussion on the criteria (ex.: limits of MIC values) that define if a preparation is active or not active, formulate your own criteria and discuss critically the published results. Have in mid that some methods used to investigate activity are currently considered as obsolete (for instance, inhibition zone diameter is severely influenced by the ability of the active compounds to diffuse across the solid culture media and for that reason the values of "inhibition zone diameters" do not just reflect inhibition.

- Review the text for typesetting: italicize scientific names plant species.

Author Response

(The authors gave the same response as above.)

Round 2

Reviewer 1 Report

The authors have successfully addressed my comments.

Reviewer 2 Report

Thanks you for your response. The manuscript is much more better now. 

Reviewer 3 Report

Dear authors,

Thank you for considering my comments produced during the first review round.

The changes you introduced in the manuscript and your explanations regarding the raised questions are consistent and convincing, Thus if editors agree that the manuscript falls within the aim of the special issue "Ethnobotany, Medicinal Plants and Biodiversity Conservation", I recommend on its acceptance in the present form.

With kind regards,